# Measuring Attitudes Toward Plastics: A Cross-Cultural Adaptation and Patient Evaluation Study

**DOI:** 10.3390/ijerph22121857

**Published:** 2025-12-12

**Authors:** Francesca Diodati, Denisa Gabriela Balan, Giovanni Libralato, Loredana Manfra, Valerio Vanelli, Matteo Puntoni, Caterina Caminiti

**Affiliations:** 1Clinical and Epidemiological Research Unit, University Hospital of Parma, 43126 Parma, Italy; fdiodati@ao.pr.it (F.D.); denisa.balan@ao.pr.it (D.G.B.); ccaminiti@ao.pr.it (C.C.); 2Department of Biology, University of Naples Federico II, 80126 Naples, Italy; giovanni.libralato@unina.it; 3Institute for Environmental Protection and Research (ISPRA), 00144 Rome, Italy; loredana.manfra@isprambiente.it; 4Department of Sociology and Business Law, University of Bologna, 40126 Bologna, Italy; valerio.vanelli@unibo.it; 5Department of Political and Social Sciences, University of Bologna, 40126 Bologna, Italy; 6Department of Communication and Economics, University of Modena and Reggio Emilia, 41121 Modena, Italy

**Keywords:** plastics, biodegradable, pollution, surveys, questionnaires, cross-cultural adaptation, citizens, knowledge, awareness, behavior-change interventions

## Abstract

Individual behaviors play a crucial role in generating and mitigating plastic pollution. Understanding citizen knowledge and perceptions is therefore critical to inform effective public interventions. Surveys can provide this information, but they must use well-designed and culturally adapted tools to be reliable. We present the Italian cross-cultural adaptation of an Australian questionnaire identified via systematic review as meeting high-quality standards. The tool included 21 items (Likert-scale, multiple-choice, and open-ended). In accordance with literature indications, we performed forward and back translation and subsequent review by an Expert Committee, producing a pre-final version. A stratified sample of 43 citizens assessed clarity of each item and provided feedback, which guided further Expert Committee revision. Ten items showed comprehension problems, and seven of them were rephrased because they were confusing or redundant. Items with technical terms such as “bioplastics” and “biodegradable” proved challenging, leading to the addition of brief explanations in the introduction to the questionnaire. This process produced a rigorously developed, culturally appropriate instrument for assessing public understanding of plastic pollution in Italy. This standardized tool, if adapted in multiple languages, will enable international surveys and meta-analyses to guide global strategies. Psychometric validation is recommended before large-scale deployment of the tool.

## 1. Introduction

Plastic pollution has emerged as one of the most pressing environmental challenges of our time, posing threats to both planetary and human health [1,2,3]. There is no single or universal solution to this complex problem. Reducing primary plastic production and use will be necessary [4], yet a world entirely without plastics is neither conceivable [5] nor desirable, given their essential role in economic growth and potential contribution to lower-carbon solutions [6]. Recycling is widely promoted as a priority measure, but current global recycling rates remain below 10% [5,7]. Innovative degradable and bio-based plastics offer promising alternatives, although uncertainties persist regarding their true benefits and possible harms [8,9]. Tackling plastic pollution therefore requires a multifaceted approach, engaging all stakeholders-policymakers, producers, and end-users—in coordinated action [10].

Among these stakeholders, the responsibility and potential impact of individual citizens are often underestimated [10]. Yet, rational use of plastics and appropriate waste disposal are essential, and individual behaviors are a key driver of plastic litter generation [11,12]. In fact, evidence suggests that changes in consumer practices could substantially contribute to alleviating plastic pollution, especially when realized alongside regulatory action [12]. Understanding public attitudes and awareness—regarding plastics and alternative materials, perceived risks, and acceptance of available solutions—is thus critical for designing effective strategies [13,14]. Identifying gaps in knowledge or support can improve communication about policy measures, while insights into consumer perceptions can also inform producers about potential market acceptance of alternative products [15]. Because such information is inherently subjective, it can only be obtained through self-report surveys [16].

However, surveys can only be reliable and informative if they use rigorously developed and validated questionnaires [17], and are adapted to the language and culture of the people being surveyed [18,19,20,21]. Consequently, selecting the appropriate instruments for a given survey is a complex task, which must include careful quality assessment [22]. In a recent systematic review, we analyzed survey studies measuring public knowledge and awareness about plastics, plastic pollution, and mitigation strategies [23]. Using the Burns and Kho assessment guide [17] and the AXIS appraisal tool for cross-sectional studies [24], we found that most studies lacked rigorous methodology and did not employ validated instruments. The Australian survey by Dilkes-Hoffman et al. [25,26] appeared to be the most consistent with quality standards.

To make this tool available for Italian-speaking citizens, we undertook a cross-cultural adaptation of the Australian instrument. This process is essential to ensure linguistic, cultural, and conceptual equivalence when applying a questionnaire in a different country and context [18,19,20,21]. We herein report the results of this adaptation process, conducted within the framework of BIOPLAST4SAFE [27], a national project funded by the Italian Ministry of Health.

## 2. Materials and Methods

We followed the Task Force for Translation and Cultural Adaptation of the International Society for Pharmacoeconomics and Outcomes Research (ISPOR) framework [19] and the guidelines by Beaton et al. [18] to plan, conduct, and report this study. Figure 1 summarizes the method used.

### 2.1. The Instrument

The questionnaire developed by Dilkes-Hoffman et al. comprises 21 items, reported by the authors in two separate papers: one investigating public attitudes towards plastics [25] and the other on bioplastics [26]. The tool used in the present study comprised both parts: 14 items on plastics and 7 items specific to biodegradable plastics and other alternative materials. In accordance with literature indications [17], the original instrument was created through a systematic process: items were derived from existing environmental surveys and refined via several rounds of prototyping with both researchers and members of the public. It was subsequently piloted in a random sample of 250 citizens, leading to minor modifications that were incorporated into the final version used for formal data collection.

The tool includes a mix of Likert-scale, multiple-choice, and open-ended questions. Open-ended items were positioned at the beginning of the questionnaire to reduce the risk of response bias induced by the wording and context of later items.

### 2.2. Cross-Cultural Adaptation

Cross-cultural adaptation is a structured process designed to achieve equivalence between the original source and the translated version of an instrument, accounting for linguistic, cultural, and contextual factors [18]. Although multiple methodological guidelines are available [20], they generally share common steps. In this work, we primarily referred to the principles established by the ISPOR Task Force [19] and the method proposed by Beaton et al. [18], widely adopted in cross-cultural research.

#### 2.2.1. Preparation

Before initiating the translation process, as the two papers by Dilkes-Hoffman et al. [25,26] reported on two different sets of questionnaire items, we combined the content into one single tool. We also contacted the authors prior to commencing the study and obtained permission to use the instrument.

#### 2.2.2. The Translation Process

The original English-language instrument was translated into Italian (forward translation) by a professional translator experienced in health science texts but blinded to the project’s specific objectives. The Italian version was then back translated into English by a similarly blinded, certified native-English translator with expertise in scientific writing. Both professionals provided a report documenting key challenges and justifications for terminology choices.

#### 2.2.3. Expert Review

An Expert Committee (EC) was established, including the two translators, a methodologist, a linguist, two environmental researchers, and a sociologist with expertise in survey methodology. The EC received an item history table containing the original items, the translations, and the translators’ comments.

As a first step, members were asked to rate the relevance of each questionnaire item for the Italian context and project objectives on a 5-point Likert scale (1 = not relevant, 5 = very relevant) [20]. Items receiving low ratings would be flagged for possible removal. Subsequently, the experts met online to review all items in detail, focusing on discrepancies and critical issues. Following Beaton’s recommendations, the review process specifically examined four dimensions of equivalence: semantic, idiomatic, experiential, and conceptual. This process resulted in a pre-final Italian version of the questionnaire.

#### 2.2.4. Pretest (Cognitive Debriefing)

The pre-final version was administered online, using the REDCap platform [28,29], to a sample of Italian-speaking citizens to assess clarity and comprehension. Members of the research team invited potential participants via email and further recruitment was achieved through snowball sampling. Participation was voluntary and uncompensated. Consistent with published recommendations [18,20], a target sample size of 30–40 respondents was set. To enhance external validity, stratified recruitment criteria were adopted based on gender, age group, education level, and occupational status [19]. Moreover, 10% of participants was required to be of non-Italian nationality. Proportions for each stratum were defined by the consulting sociologist, using Italian population statistics as a reference [30]. Details on the sample distribution plan are provided in Appendix A. Anonymity was guaranteed as no identifiable personal data were collected, but respondents were invited to contact the research team for further information or to be updated on the study results.

Participants completed the questionnaire and rated the clarity of each item and response option on a 3-point scale (1 = “it is not clear”, 2 = “item needs some revision”, 3 = “very clear”) [20]. For ratings of 1 or 2, respondents were prompted to provide suggestions for improvement.

A summary report of pretest results was shared with the EC, which reconvened online to review the findings. Based on participant ratings and comments, further adjustments were made, leading to the finalization of the Italian version of the questionnaire.

### 2.3. Statistical Analysis

We included in the analysis the questionnaires where at least 80% of the items were completed. Descriptive statistics were applied to summarize the characteristics of participants in the pretest sample. Categorical variables (gender, age group, education level, occupational status, nationality) were reported as absolute frequencies and percentages, and displayed in contingency tables and bar charts.

For each questionnaire item, the clarity ratings were summarized as frequencies and proportions. The distribution of responses was graphically illustrated using bar charts and histograms, and the proportion of items rated as “very clear” was calculated overall and stratified by participant subgroups.

Qualitative comments and suggestions provided by participants were collated and analyzed thematically to identify recurrent issues. These findings were used to guide subsequent revisions of the questionnaire.

All analyses were conducted using R 4.4.0.

## 3. Results

The study formally began on 19 November 2024, with the original questionnaire being submitted to the translators, and ended on 22 April 2025 with the enrolment of the last subject. Presented below are the outcomes of expert review, which resulted in the drafting of the pre-final version of the questionnaire, and of the pretest (cognitive debriefing), which led to finalization of the Italian version.

### 3.1. Expert Review and Definition of the Pre-Final Version

In their preliminary assessment, the experts rated all items as relevant (scores of 3 or 4), therefore all items were maintained for cross-cultural adaptation. Subsequently, the EC met via videoconference on 2 December 2024. The comparison of the backward translation against the original did not highlight important discrepancies, indicating that the forward translation was substantially accurate. However, the careful review of the translations raised a few relevant issues that were thoroughly discussed and led to unanimous decisions. As a rule, it was agreed to make the text as simple and straightforward as possible for Italian speakers to enhance comprehension. This required changes in sentence structure, such as converting indirect questions to direct ones, and using the first person instead of the second. This was judged to increase clarity and immediacy, while maintaining semantic equivalence. Along these lines, to facilitate questionnaire completion, five items were modified to ensure coherence and uniformity within the questionnaire or to remove excessive descriptions. Also, various lexical changes were introduced (six items) to make the meaning more explicit for Italian-speaking people, thus ensuring experiential equivalence. Examples are the translation of “ocean” with the Italian for “sea”, reference to “Italy” instead of “Australia”, and the paraphrasing of “landfill” and “composting”, as these are uncommon waste management practices in Italy.

Some adjustments were also made to response scales, which were standardized to a 5-point metric for consistency. Also, for clarity, an explanatory term was added to all response options, whereas in the original sometimes only a number or a percentage was given.

### 3.2. Pretest (Cognitive Debriefing)

The questionnaire was uploaded to the REDCap platform, hosted at the University Hospital of Parma. A total of 44 questionnaires were submitted, of which one was excluded as fewer than 80% of items had been completed. Thus, 43 questionnaires were analyzed. General demographic characteristics of participants are provided in Table 1.

Age ranged from 19 to 82 years (median 52), with equal representation of males and females. All subjects had at least a middle school degree, with the majority (41%) indicating “high school” as their highest level of education. Most respondents were currently employed (60.5%), followed by an equal proportion of retirees and students (9.3%). Compared with Italian statistics [30], the characteristics of our sample closely match those of the Italian population, save for a smaller proportion of elderly retirees, due to the difficulty we encountered in identifying interested subjects willing to take part in our survey in this age class.

Participant assessments of clarity are detailed in Appendix A. Overall, 11/21 (52.3%) items were rated as “very clear” by over 95% of respondents. Of the 10 items exhibiting clarity problems (Figure 2), 4/14 (28.6%) concerned plastics (D2, D7, D9, D10), and 6/7 (85.7%) were related to bioplastics and other materials (D12, D13, D14, D15, D17, D18).

#### Analysis of Responses and Definition of the Final Italian Questionnaire

The EC met again on 16 June 2025, to review the responses and comments provided by citizens and discuss any necessary improvements. Of the ten items rated as “unclear” or “needing some revision” (Figure 2), seven were judged to use confusing, ambiguous or redundant wording, and were consequently rephrased to facilitate comprehension and completion, without affecting semantic equivalence.

Specifically, the discussion of the EC focused on three main aspects. First, the response options “Harmful/Beneficial” and “Bad/Good”, included in three items (D7, D9, D13), were found to repeat the same concept covered by other response options “Bad/Good for the environment” and “Bad/Good for my health”. Therefore, the EC decided to remove options to improve ease of completion, still preserving equivalence. Second, following citizen feedback, the EC rephrased Item D10, which explored the level of agreement with three statements indicating actions to counter plastic pollution. The EC chose to ask a more direct question investigating willingness to perform such actions. Third, the greatest challenges in comprehension were observed for the seven items containing technical terms indicating alternative materials: degradable, biodegradable, and bio-based (D12–D18). On this point, the EC debated whether it would be appropriate to add an explanation of these terms. Since the items of the questionnaire do not directly investigate knowledge of different types of plastics, the EC decided to leave these three items unchanged, but to add short explanations of the three terms to the introduction of the tool.

Following the meeting, the revised questionnaire incorporating all agreed changes was circulated to EC members, who subsequently approved it as final (Appendix A).

## 4. Discussion

Choosing the right questionnaire to conduct surveys can be challenging [22]. To generate reliable results and provide useful insights, surveys must use rigorously developed instruments, appropriate to the linguistic, cultural, and social context of the target population [19,22]. With this work, we make available an appropriate questionnaire to measure knowledge and awareness of plastic pollution in Italian-speaking people. The information collected with this tool can help identify behavioral strategies that may work in Italian settings, and more generally in Italian-speaking citizens all over the world. This, in turn, can inform targeted, culturally appropriate behavior-change interventions to reduce plastic consumption, improve recycling practices, and limit littering [12]. Furthermore, this tool, when appropriately translated, could be used in international surveys to enable data comparison and highlight national peculiarities that require priority interventions. The scientific contribution of this work therefore extends beyond the Italian context.

This study has strengths. Notably, the tool we selected for linguistic validation was developed on solid methodological grounds by a well-established research group that continues to refine it through constant research [15]. Consent from the original authors was obtained prior to initiating the study, ensuring methodological alignment and fidelity to the source instrument. Another major asset of our study is the rigor of the cross-cultural adaptation process, which included the contribution of a multidisciplinary group of experts and, above all, the active participation of citizens whose characteristics reflected those of the Italian population. Public input not only helped improve the clarity of the questionnaire, but also highlighted challenging themes in the area of plastic pollution deserving particular attention. Specifically, items containing terms defining alternative materials (e.g., biodegradable, or bio-based), were found to be problematic by a high number of respondents. Confusion and comprehension difficulties in relation to these terms have already been reported in the literature [31] and are also emphasized in the recent survey by Dilkes-Hoffman et al. [15], who call for further research to understand the impact of misleading messaging and other potential drivers behind the observed knowledge gaps.

Some limitations should also be acknowledged. Firstly, the number of participants in the pretest was relatively small, which may have restricted the diversity of feedback obtained. Also, recruiting older participants proved challenging, leading to an underrepresentation of this age group with respect to Italian population statistics. This may have been due to the choice of internet administration, as research shows that older persons often exhibit a preference for traditional modes of communication, such as mail or face-to-face interactions [32]. Furthermore, the literature does not indicate that the lower participation of older citizens may be driven by a lack of interest in the topic, since older generations often display stronger pro-environmental attitudes than younger people [33]. The under-representation of the elderly may have influenced overall assessments leading to an overestimation of clarity, since older adults may process or interpret certain terms or concepts differently than younger individuals and have varying perceptions regarding plastics. These issues were somewhat mitigated by the EC’s careful attention to the observations raised by older respondents, ensuring that their input was fully considered. However, mixed-mode administration approaches, with the option to use a physical questionnaire, combined with appropriate recruitment strategies should be considered when applying this tool in population studies [34]. Secondly, we did not carry out formal psychometric testing (evaluation of validity, reliability, responsiveness), which is necessary to establish the questionnaire’s measurement properties and allow for confident interpretation of results [18,22,35]. Notably, to the best of our knowledge, psychometric testing of the original Australian questionnaire has not yet been reported. If resources allow, we plan to conduct a full validation of the Italian version, ideally in collaboration with the Australian team in a cross-national study. Particularly relevant will be the assessment of responsiveness, i.e., the ability of the instrument to detect changes over time, since understanding how people’s knowledge and perceptions evolve in response to political, social, and contextual developments is essential to plan targeted interventions promoting pro-environmental behavior.

## 5. Conclusions

Through a rigorous adaptation process, this study provides a questionnaire that can be used in Italian-speaking citizens to assess public attitudes and behaviors related to plastic use. It can therefore be a valuable resource for policymakers and researchers in the field of public health for designing campaigns and interventions to counter plastic pollution. Importantly, a standardized instrument, if available in multiple languages, will enable international, large-scale surveys and the use of results in meta-analyses to guide global strategies.

## Figures and Tables

**Figure 1 ijerph-22-01857-f001:**
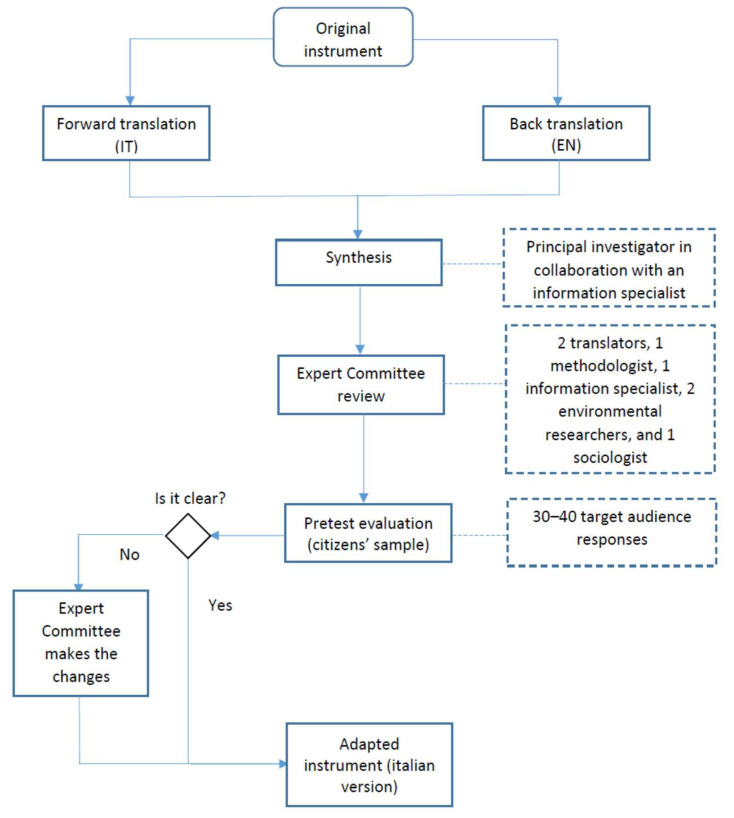
Process of translation and cross-cultural adaptation of the questionnaire (based on ISPOR/Beaton guidelines).

**Figure 2 ijerph-22-01857-f002:**
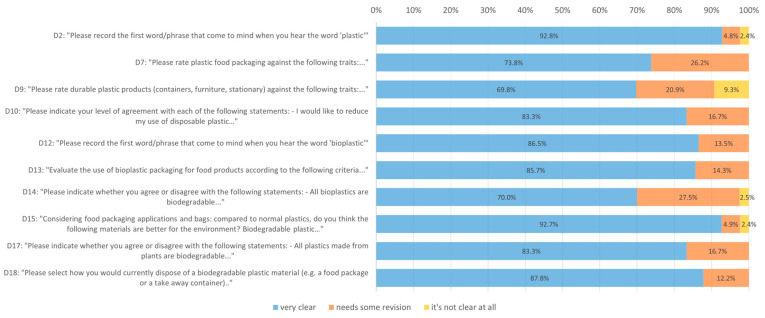
Bar chart showing the frequency of responses to the 10 questions that showed clarity issues. Note: for each item, the color of the bar identifies the percentage of respondents who reported the need for a thorough re-evaluation (yellow), who required some adjustment (orange) and who expressed complete clarity (blue).

**Table 1 ijerph-22-01857-t001:** Socio–demographic and occupational characteristics of respondents (N = 43).

Item	Value	Italian Population Statistics *
**Age**, median (min–max)	52.0 (20–82)	
20–29 years	6 (14.0%)	12.3%
30–49 years	9 (20.9%)	29.3%
50–64 years	20 (46.5%)	28.6%
65–74 years	2 (4.7%)	14.3%
75+ years	3 (7.0%)	15.5%
No response	3 (7.0%)	
**Sex**		
Female	21 (48.8%)	51.5%
Male	21 (48.8%)	48.5%
No response	1 (2.3%)	
**Highest education level**		
Middle school	15 (34.9%)	44.9%
High school	18 (41.9%)	47.1%
University/Postgraduate	10 (23.3%)	8.0%
**Employment status**		
Currently employed	26 (60.5%)	48.8%
Retired	4 (9.3%)	33.3%
Student	4 (9.3%)	3.8%
Other	3 (7%)	14.1%
No response	6 (14.0%)	
**If employed: job type**		
Manager, entrepreneur, self-employed professional	1 (3.8%)	6.3%
Executive, middle manager, employee, worker/apprenticeship	14 (53.8%)	66.3%
Self-employed worker, contributing family worker, freelance	3 (11.5%)	27.4%
No response	8 (30.8%)	

Note: * Data on age, Sex, Employment status, Job Type refer to the first trimester of 2025. Highest education level updated on 1 January 2020. The comparison enables to identify differences between the characteristics of the study sample and those of the Italian population according to official statistics [30].

## Data Availability

All data generated or analyzed during this study are available from the authors.

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
