# Peer review of "Measuring Attitudes Toward Plastics: A Cross-Cultural Adaptation and Patient Evaluation Study"

_ijerph, 2025, doi:10.3390/ijerph22121857_

Round 1
Reviewer 1 Report
Comments and Suggestions for Authors
This study presents a cross-cultural adaptation of an Australian questionnaire to measure public attitudes toward plastic pollution in Italy. While the research is valuable and innovative for the Italian context, there are some minor issues that need to be addressed.
1.The questionnaire contains several technical terms, particularly around biodegradable and bioplastics, which seem to cause confusion among respondents. While I understand the decision not to provide definitions to avoid bias, it may be helpful to consider adding brief explanations in the introduction or other section to ensure that respondents have a clear understanding of these terms.
2.In a few items, such as D7, D9, and D13, there is some redundancy in the response options. While the changes made to eliminate this redundancy are a step in the right direction, I suggest revisiting other similar items to ensure that the response options are as distinct and non-overlapping as possible.
3. The translation of terms was adjusted to better reflect Italian waste management practices, which is a good step. However, further attention could be given to the examples used for bioplastics. In Italy, the public’s familiarity with these alternatives might differ significantly from other regions. It could be beneficial to add examples of local bioplastic products or packaging in the questionnaire.
4. While the pretest sample includes a range of age groups, I noticed that elderly participants were underrepresented. This is important as older adults might have different perceptions and behaviors regarding plastics.
5. Some of the Likert-scale questions related to environmental impact, like D7and D9, might benefit from more detailed explanation to clarify what "Good/Bad for the environment".
Author Response
1.The questionnaire contains several technical terms, particularly around biodegradable and bioplastics, which seem to cause confusion among respondents. While I understand the decision not to provide definitions to avoid bias, it may be helpful to consider adding brief explanations in the introduction or other section to ensure that respondents have a clear understanding of these terms.
This issue was indeed the subject of considerable debate within the Expert Committee, given the comprehension difficulties reported by many respondents. In the light of the observations received from the reviewer, also raised by reviewer 3, the Expert Committee has decided to include in the introduction to the questionnaire brief explanations of the technical terms. The items of the questionnaire were instead left unchanged to maintain consistency with the original version. The relevant parts of the abstract and main text (Results section 3.2.1) have been revised to reflect this change.
- In a few items, such as D7, D9, and D13, there is some redundancy in the response options. While the changes made to eliminate this redundancy are a step in the right direction, I suggest revisiting other similar items to ensure that the response options are as distinct and non-overlapping as possible.
In consideration of this comment, the Expert Committee has carefully reviewed all items, and did not identify any further redundant response options. Since the adapted version should adhere to the original as much as possible, we believe no further changes should be introduced.
- The translation of terms was adjusted to better reflect Italian waste management practices, which is a good step. However, further attention could be given to the examples used for bioplastics. In Italy, the public’s familiarity with these alternatives might differ significantly from other regions. It could be beneficial to add examples of local bioplastic products or packaging in the questionnaire.
As indicated in response 1, we have now added explanatory notes to the introduction of the questionnaire. These include a few examples of commonly found products made of these materials.
- While the pretest sample includes a range of age groups, I noticed that elderly participants were underrepresented. This is important as older adults might have different perceptions and behaviors regarding plastics.
We agree this is an important aspect. We have now elaborated on this issue in the Limitations, providing possible explanations and hypothesizing the impact it may have on our results.
- Some of the Likert-scale questions related to environmental impact, like D7and D9, might benefit from detailed explanation to clarify what "Good/Bad for the environment".
We agree that the terms “good” and “bad” may sound generic. However, the use of these colloquial and direct dichotomous expressions may facilitate immediacy of judgment from the respondents. Since this was a choice made by the authors of the original questionnaire, we believe the Italian version should adhere to the English wording.
Reviewer 2 Report
Comments and Suggestions for Authors
Dear Authors,
I have reviewed your manuscript entitled “Measuring Attitudes Toward Plastics: A Cross-Cultural Adaptation and Patient Evaluation Study.”
I enjoy reading your manuscript.
The topic is highly relevant and timely, addressing a critical methodological gap in assessing public attitudes and knowledge toward plastics and plastic pollution.
It is clear that the authors have made an effort to write a good article with solid references. (ISPOR, Beaton et al.), and the process is described with commendable rigor and transparency
My review will be done by chapters:
Abstract: The abstract is concise and informative, but it could benefit from a clearer statement on the implications of this adaptation. For instance, specify how this tool might be used to inform national policies or international comparisons
Introduction
The introduction is comprehensive and well-referenced.
Results:
Very well structured: highlights strengths, limitations, and prospects. Suggestion: Reinforce the scientific contribution and international applicability (this increases the perceived impact).
Small adjustment: “psychometric testing has not yet been performed on the original Australian questionnaire” → great point, but could be more diplomatic, e.g.: “to our knowledge, psychometric testing has not yet been reported.”
Conclusion:
The conclusion fulfills its basic function, but could be expanded and refined to sound more academic, assertive, and consistent with the tone of the rest of the article.
I would suggest the following to the authors:
Briefly summarize the objective and main results.
Emphasize the scientific and practical contribution of the study;
Point to future directions (without repeating the discussion).
Author Response
Abstract:
The abstract is concise and informative, but it could benefit from a clearer statement on the implications of this adaptation. For instance, specify how this tool might be used to inform national policies or international comparisons
We agree these are important points that need to be stressed, also in the abstract. We have rephrased parts of the abstract to emphasize the implications of our work, including the possible future applications of this tool.
Results:
Very well structured: highlights strengths, limitations, and prospects. Suggestion: Reinforce the scientific contribution and international applicability (this increases the perceived impact).
We thank the reviewer for this suggestion that has enabled us to emphasize the value of this work. We have now enriched the discussion accordingly.
Small adjustment: “psychometric testing has not yet been performed on the original Australian questionnaire” → great point, but could be more diplomatic, e.g.: “to our knowledge, psychometric testing has not yet been reported.”
We agree. We have rephrased as suggested
Conclusion:
The conclusion fulfills its basic function, but could be expanded and refined to sound more academic, assertive, and consistent with the tone of the rest of the article. I would suggest the following to the authors: Briefly summarize the objective and main results. Emphasize the scientific and practical contribution of the study; Point to future directions (without repeating the discussion).
We have improved the Conclusions as suggested.
Reviewer 3 Report
Comments and Suggestions for Authors
The study is relevant, interesting, and clearly important for understanding public attitudes toward plastics. The overall structure of the paper is good, and the methodology follows recognized guidelines.
The introduction gives solid background, but adding a few more recent references about cross-cultural adaptation and questionnaire validation would make it stronger and help readers understand why your approach is necessary.
The description of the sample is helpful, but it would be clearer if you explain how the proportions for each stratum were decided and why it was difficult to recruit older participants. A short explanation of how this might influence the results would improve transparency.
The Expert Committee process is described well, but including one short example of how a specific item was discussed or modified would help readers better understand the decision-making process.
Several technical terms—such as “biodegradable,” “bio-based,” and “degradable”—were confusing for many participants. The choice not to modify these items is reasonable, but it would be beneficial to explain this decision more directly, especially how providing definitions might bias the responses.
The paper mentions that psychometric testing has not been done. Adding a short statement about the planned next steps (e.g., reliability testing, factor analysis, responsiveness) would help show the future development of the tool.
Some sentences in the manuscript are long and could be made shorter to improve clarity and flow. A few references also seem to need small formatting fixes.
The figures and tables are clear, but adding small clarifying notes in the captions could help guide readers on what to pay attention to.
While the study is solid, there are several points should be aware of. These are not major flaws, but they do deserve attention to ensure clarity and impact.
First, the English language is generally clear but would benefit from light editing for readability. For example, sentences such as “To be reliable and informative, however, surveys must be based on rigorously developed and validated questionnaires” are grammatically correct but a bit heavy and could be simplified to improve flow. Likewise, long sentences with multiple clauses-e.g., “Some adjustments were also made to response scales, which were all presented on a 5-point metric for uniformity”—may be challenging for some readers. A minor language edit would enhance clarity.
Second, the pretest sample is relatively small (N = 43) and underrepresents older adults. The authors acknowledge this, but it would be beneficial if they clarified more explicitly how this might influence the clarity ratings or the generalizability of the pre-final version. Recruitment challenges are understandable, yet a more transparent discussion would strengthen the study’s trustworthiness.
Third, the study stops at cross-cultural adaptation and does not include psychometric validation. This is not a methodological flaw but a limitation that affects the tool’s readiness for large-scale research. The authors briefly mention this, but it may be helpful to encourage them to outline concrete validation plans (e.g., reliability testing, factor analysis, and responsiveness) to give readers a clearer sense of next steps. Given the growing expectations in the field, this addition could enhance the manuscript’s perceived robustness.
Another point worth noting is the predictable difficulty participants had with technical terms such as “biodegradable,” “bio-based,” and “degradable.” This is not a weakness in the authors’ work, but it highlights an important insight: even in an educated sample, these concepts remain unclear. The authors correctly chose not to modify the items to avoid biasing responses, but they could better emphasize this methodological justification to prevent misunderstandings among reviewers or readers.
Overall, the manuscript makes a meaningful contribution. With these improvements in clarity, explanation, and structure, the paper will be even stronger and easier for readers to follow.
Comments on the Quality of English LanguageThe English in the manuscript is generally understandable, but there are several places where the writing can be smoother and clearer. Some sentences are long and contain too many ideas at once, which makes them harder to read. For example, the sentence “Surveys can capture this information, but their results will only be reliable if they employ instruments that are rigorously designed and culturally adapted” can be written more simply as “Surveys can provide this information, but they must use well-designed and culturally adapted tools to be reliable.” There are also sentences that sound overly formal and could flow more naturally, such as “This process resulted in a rigorously developed, culturally appropriate instrument,” which could be simplified to “This process produced a clear and appropriate tool for the Italian context.” In addition, terms like “bioplastics,” “biodegradable,” and “bio-based” appear without explanation and may confuse readers, especially in long sentences where the structure becomes heavy. Shortening these sentences and keeping terminology consistent will improve readability. Another example is “Ten items exhibited comprehension issues. Seven were judged redundant or confusing and were consequently rephrased,” which could be smoothed into “Ten items showed comprehension problems, and seven of them were rephrased because they were confusing or redundant.” Overall, the English is acceptable, but a light language edit focusing on making sentences shorter, improving transitions, and simplifying complex phrasing will make the manuscript clearer and easier to follow.
Author Response
The introduction gives solid background, but adding a few more recent references about cross-cultural adaptation and questionnaire validation would make it stronger and help readers understand why your approach is necessary.
We have added references to recent papers on cross-cultural adaptation and questionnaire validation in the Introduction.
The description of the sample is helpful, but it would be clearer if you explain how the proportions for each stratum were decided and why it was difficult to recruit older participants. A short explanation of how this might influence the results would improve transparency.
We agree these are very important points that deserve elaboration.
The composition of the sample was defined based on a thorough analysis by the expert sociologist. To make choices transparent, we have now provided a detailed description of this work in the supplementary material.
Regarding the lower participation of older participants, we have expanded on this aspect in the Limitations, providing possible explanations and hypothesizing the impact it may have on our results.
The Expert Committee process is described well, but including one short example of how a specific item was discussed or modified would help readers better understand the decision-making process.
We are grateful to the reviewer for this suggestion. We have detailed in the Results section (Paragraph 3.2.1) the decision process for three main issues debated by the Expert Committee. This will help reader understand the process, and also emphasize the rigor of our work.
Several technical terms—such as “biodegradable,” “bio-based,” and “degradable”—were confusing for many participants. The choice not to modify these items is reasonable, but it would be beneficial to explain this decision more directly, especially how providing definitions might bias the responses.
This issue was indeed the subject of considerable debate within the Expert Committee, given the comprehension difficulties reported by many respondents. In the light of the observations received from the reviewer, also raised by reviewer 1, the Expert Committee has decided to include in the introduction to the questionnaire a brief explanation of the technical terms. The items of the questionnaire were instead left unchanged to maintain consistency with the original version. The relevant parts of the abstract and main text (Results section 3.2.1) have been revised to reflect this change.
The paper mentions that psychometric testing has not been done. Adding a short statement about the planned next steps (e.g., reliability testing, factor analysis, responsiveness) would help show the future development of the tool.
This is actually outlined in the Limitations section: “If resources allow, we plan to conduct a full validation of the Italian version, ideally in collaboration with the Australian team in a cross-national study. Particularly relevant will be the assessment of responsiveness, i.e., the ability of the instrument to detect changes over time, since understanding how people’s knowledge and perceptions evolve in response to political, social, and contextual developments is essential to plan targeted interventions promoting pro-environmental behavior.”
Some sentences in the manuscript are long and could be made shorter to improve clarity and flow. A few references also seem to need small formatting fixes.
We thank the reviewer for the suggestions related to language and syntax which have enabled us to improve the readability and flow of our manuscript. We have made the suggested changes, simplified various sentences and streamlined the text.
The figures and tables are clear, but adding small clarifying notes in the captions could help guide readers on what to pay attention to.
We agree. We have added explanatory notes to tables and figures
First, the English language is generally clear but would benefit from light editing for readability. For example, sentences such as “To be reliable and informative, however, surveys must be based on rigorously developed and validated questionnaires” are grammatically correct but a bit heavy and could be simplified to improve flow. Likewise, long sentences with multiple clauses-e.g., “Some adjustments were also made to response scales, which were all presented on a 5-point metric for uniformity”—may be challenging for some readers. A minor language edit would enhance clarity.
Please see response above.
Second, the pretest sample is relatively small (N = 43) and underrepresents older adults. The authors acknowledge this, but it would be beneficial if they clarified more explicitly how this might influence the clarity ratings or the generalizability of the pre-final version. Recruitment challenges are understandable, yet a more transparent discussion would strengthen the study’s trustworthiness.
Please see response above.
Third, the study stops at cross-cultural adaptation and does not include psychometric validation. This is not a methodological flaw but a limitation that affects the tool’s readiness for large-scale research. The authors briefly mention this, but it may be helpful to encourage them to outline concrete validation plans (e.g., reliability testing, factor analysis, and responsiveness) to give readers a clearer sense of next steps. Given the growing expectations in the field, this addition could enhance the manuscript’s perceived robustness.
Please see response above.
Another point worth noting is the predictable difficulty participants had with technical terms such as “biodegradable,” “bio-based,” and “degradable.” This is not a weakness in the authors’ work, but it highlights an important insight: even in an educated sample, these concepts remain unclear. The authors correctly chose not to modify the items to avoid biasing responses, but they could better emphasize this methodological justification to prevent misunderstandings among reviewers or readers.
Please see response above.
Comments on the Quality of English Language
The English in the manuscript is generally understandable, but there are several places where the writing can be smoother and clearer. Some sentences are long and contain too many ideas at once, which makes them harder to read. For example, the sentence “Surveys can capture this information, but their results will only be reliable if they employ instruments that are rigorously designed and culturally adapted” can be written more simply as “Surveys can provide this information, but they must use well-designed and culturally adapted tools to be reliable.” There are also sentences that sound overly formal and could flow more naturally, such as “This process resulted in a rigorously developed, culturally appropriate instrument,” which could be simplified to “This process produced a clear and appropriate tool for the Italian context.”. In addition, terms like “bioplastics,” “biodegradable,” and “bio-based” appear without explanation and may confuse readers, especially in long sentences where the structure becomes heavy. Shortening these sentences and keeping terminology consistent will improve readability. Another example is “Ten items exhibited comprehension issues. Seven were judged redundant or confusing and were consequently rephrased,” which could be smoothed into “Ten items showed comprehension problems, and seven of them were rephrased because they were confusing or redundant.” Overall, the English is acceptable, but a light language edit focusing on making sentences shorter, improving transitions, and simplifying complex phrasing will make the manuscript clearer and easier to follow.
Please see response above